Static and dynamic alterations in the amplitude of low-frequency fluctuation in patients with amyotrophic lateral sclerosis

Ma Xujing 1
Lu Fengmei 2 3
Chen Heng 4
Hu Caihong 1
Wang Jiao 1
Zhang Sheng 1
Zhang Shuqin 1
Yang Guiran yangguiran2006@126.com 1
Zhang Jiuquan zhangjq_radiol@foxmail.com 5 6 7 8
1 Department of Medical Technology, Cangzhou Medical College , Cangzhou , China
2 The Clinical Hospital of Chengdu Brain Science Institute , Chengdu , China
3 MOE Key Lab for Neuroinformation, School of life Science and Technology, University of Electronic Science and Technology of China , Chengdu , China
4 School of Medicine, Guizhou University , Guiyang , China
5 Department of Radiology, Chongqing University Cancer Hospital , Chongqing , China
6 Key Laboratory for Biorheological Science and Technology of Ministry of Education, Chongqing University , Chongqing , China
7 Chongqing Cancer Institute , Chongqing , China
8 Chongqing Cancer Hospital , Chongqing , China
Liu Feng
Electronic publication date: 2020 Nov 2
Publication date: 2020
Volume: 8
Electronic Location ID: e10052
Received 2020 Jun 12; Accepted 2020 Sep 7
Copyright: ©2020 Ma et al.
Copyright year: 2020
Copyright holder: Ma et al.
License: This is an open access article distributed under the terms of the Creative Commons Attribution License, which permits unrestricted use, distribution, reproduction and adaptation in any medium and for any purpose provided that it is properly attributed. For attribution, the original author(s), title, publication source (PeerJ) and either DOI or URL of the article must be cited.
License URL: https://creativecommons.org/licenses/by/4.0/

Keywords: Amyotrophic lateral sclerosis, Amplitude of low-frequency fluctuations, Dynamic, Resting state, Static

Funding: Cangzhou Science and Technology Research and Development Project 162302137 China Postdoctoral Science Foundation Grant 2019M653383 The work was supported by the Cangzhou Science and Technology Research and Development Project (No. 162302137) and the China Postdoctoral Science Foundation Grant (No. 2019M653383). The funders had no role in study design, data collection and analysis, decision to publish, or preparation of the manuscript.

==============================
Background

Static changes in local brain activity in patients suffering from amyotrophic lateral sclerosis (ALS) have been studied. However, the dynamic characteristics of local brain activity are poorly understood. Whether dynamic alterations could differentiate patients with ALS from healthy controls (HCs) remains unclear.

Methods

A total of 54 patients with ALS (mean age = 48.71 years, male/female = 36/18) and 54 (mean age = 48.30 years, male/female = 36/18) HCs underwent magnetic resonance imaging scans. To depict static alterations in cortical activity, amplitude of low-frequency fluctuations (ALFF) which measures the total power of regional activity was computed. Dynamic ALFF (d-ALFF) from all subjects was calculated using a sliding-window approach. Statistical differences in ALFF and d-ALFF between both groups were used as features to explore whether they could differentiate ALS from HC through support vector machine method.

Results

In contrast with HCs, patients with ALS displayed increased ALFF in the right inferior temporal gyrus and bilateral frontal gyrus and decreased ALFF in the left middle occipital gyrus and left precentral gyrus. Furthermore, patients with ALS demonstrated lower d-ALFF in widespread regions, including the right lingual gyrus, left superior temporal gyrus, bilateral precentral gyrus, and left paracentral lobule by comparison with HCs. In addition, the ALFF in the left superior orbitofrontal gyrus had a tendency of correlation with ALSFRS-R score and disease progression rate. The classification performance in distinguishing ALS was higher with both features of ALFF and d-ALFF than that with a single approach.

Conclusions

Decreased dynamic brain activity in the precentral gyrus, paracentral gyrus, lingual gyrus, and temporal regions was found in the ALS group. The combined ALFF and d-ALFF could distinguish ALS from HCs with a higher accuracy than ALFF and d-ALFF alone. These findings may provide important evidence for understanding the neuropathology underlying ALS.

Introduction

Amyotrophic lateral sclerosis (ALS) is a devastating disease which involves dysfunctions in movement and cognition (Hardiman et al., 2017; Van Es et al., 2017). Patients with ALS usually died within 3–5 years after symptoms appear (Roth & Shacka, 2009). At present, the therapeutic options for ALS are limited. Nevertheless, increasing lines of evidence demonstrate that early diagnosis is important for selecting available pharmacologic therapy and that appropriate palliative care has an active influence on patients’ living quality and survival (Bourke et al., 2006; Volanti et al., 2011). Now the diagnosis of ALS is still clinical, and a pronounced delay exists between the onset of symptoms and diagnosis, possibly beyond the therapeutic window (Turner et al., 2009). Timely and accurate diagnosis of ALS is urgently needed to date, and imaging biomarkers should be developed.

Recent studies combining functional and structural data depicted that functional alterations at resting state may precede structural changes in patients with cognitive impairments (Kawagoe, Onoda & Yamaguchi, 2019; Sun et al., 2016) and ALS (Abidi et al., 2020; Chipika et al., 2019). Neuroimaging approaches provide convenience for studying local brain activities and may facilitate expanding our understanding of early diagnosis of ALS (Huynh et al., 2016; Verstraete & Foerster, 2015). Resting-state functional magnetic resonance imaging (rs-fMRI) is an ideal instrument used to probe into cortical activities based on blood oxygenation level-dependent signals without performing variable tasks (Biswal et al., 1995). As an effective index to measure local brain activity, amplitude of low frequency fluctuation (ALFF) (Guo et al., 2012; Liu et al., 2013; Yu-Feng et al., 2007) has been extensively employed in ALS research. Using this approach, scholars have discovered that patients with ALS showed aberrant activation in the precentral gyrus, frontal gyrus, and occipital regions; this finding suggests that ALS is a disease involving many system with brain impairment spreading beyond the motor cortex (Bueno et al., 2019; Ma et al., 2016; Shen et al., 2018). In addition, the increased ALFF in the frontal lobe could be a candidate biomarker in ALS (Luo et al., 2012). However, these studies are on the strength of the hypothesis that the signal of rs-fMRI is static during scanning, ignoring the dynamic behavior of activities of people’s brains (Allen et al., 2014; Fu et al., 2017; Liu et al., 2017).

Dynamic amplitude of low-frequency fluctuation (d-ALFF), an indicator of the variance of ALFF, is an effective tool to explore brain dynamics in healthy people (Liao et al., 2019; Zou et al., 2009) and patients with neuropsychiatric disorders, including schizophrenia (Yang et al., 2019), generalized anxiety disorder (GAD) (Cui et al., 2019), and Parkinson’s disease (Zhang et al., 2019). In addition, Li et al. (2018a) discovered that in contrast to static ALFF abnormalities, d-ALFF abnormalities could predict the severity of suicidal ideation in major depressive disorders. d-ALFF may contribute more than ALFF in differentiating between patients diagnosed with GAD and normal controls (Cui et al., 2019). However, as far as we know, the dynamic signatures of ALFF have been rarely elucidated in ALS; furthermore, the performance of d-ALFF compared with ALFF in recognizing ALS patients from healthy controls (HCs) at an individual level remains poorly documented.

Motivated by previous studies, we utilized d-ALFF to detect changes of dynamic patterns of brain activity in ALS. We assumed that patients with ALS would exhibit altered d-ALFF patterns contrast to HCs, and that such changes could be used as features to distinguish them. We also hypothesized that the classification performance with altered d-ALFF as features would be comparable with that using altered static ALFF.

Materials & Methods

Subjects

In western countries, the rates of ALS is probably between 1 and 3 per 100,000 per year per person-years (Robberecht & Philips, 2013), while the exact prevalence in China remains unclear (Chen et al., 2015). ALS is more widespread in men than in women in different countries (Chen et al., 2015; Oskarsson, Gendron & Staff, 2018). Patients were employed from January 1, 2009 to December 31, 2013 in this work. Fifty-four patients diagnosed with ALS and 54 HCs matched in gender and age were enrolled from Southwest Hospital. The inclusion criteria were as follows: patients can lie down flat in the scanner for at least 40 min and receive none therapeutic interventions before participating in this study; and patients with ALS diagnosed on the basis of the revised El Escorial criteria of the World Federation of Neurology (Brooks et al., 2000). The exclusion criteria were as follows: patients diagnosed with frontotemporal dementia or other mental and neurological disorders; patients with major systemic diseases; patients with family trait of motor neuron diseases and other neurodegenerative disorders; and patients with cognitive impairment. The clinical status based on ALS Functional Rating Scale-Revised (ALSFRS-R) was obtained for each patient. Disease duration was computed from symptom onset to examination date. By using the equation: (48-ALSFRS-R score)/Disease duration (Ellis et al., 1999), rate of disease progression was achieved. Demographic and clinical information of subjects are displayed in Table 1.

Table 1 Demographic and clinical characteristics of the ALS patients and HCs.

Variables	ALS (n = 54)	HC (n = 54)	p value	
Gender (female/male)	18/36	18/36	1a	
Age (years)	48.71 ± 10.21	48.30 ± 8.74	0.82b	
Gray matter	649.56 ± 2842.17	668.23 ± 3251.82	0.08b	
Disease duration (months)	20.93 ± 21.56	–	–	
ALSFRS-R	32.56 ± 6.83	–	–	
Disease progression rate	1.28 ± 1.15	–	–	
Notes.

Values are mean ± variance.

ALS Amyotrophic Lateral Sclerosis

HC Healthy Control

ALSFRS-R ALS Functional Rating Scale-revised

Disease progression rate, (48-ALSFRS-R score)/time from symptom onset.

a The p value was obtained by Chi-square t-test.

b The p value was obtained by two-sample t-test.

The measurements of the Edinburgh Handedness Inventory indicated that all the subjects were right-handed. The medical research ethics committee of Southwest Hospital (the First Affiliated Hospital of the Third Military Medical University of the Chinese People’s Liberation Army) authorized this study to proceed. Informed consent from each participant was collected.

Data acquisition

Data were collected as described in our former research (Ma et al., 2015). The following parameters were used in collecting functional data: echo time (TE) = 30 ms, repetition time (TR) = 2,000 ms, flip angle (FA) = 90°, 36 slices, 1 mm gap, field of view (FOV) = 192 mm × 192 mm, thickness = 3 mm, matrix size = 64 × 64 and voxel size = 3 mm × 3 mm × 3 mm. Two hundred and forty volumes were collected for each subject. T1-weighted structural data was gathered using the following settings: TE = 2.52 ms, TR = 1,900 ms, FA = 9°, slice thickness = 1 mm, 176 slices, 0 mm gap, FOV = 256 mm × 256 mm, matrix size = 256 × 256, and voxel size = 1 mm × 1 mm × 1 mm.

Data analysis

Preprocessing

fMRI data preprocessing was performed with the Data Processing Assistant for Resting-state fMRI (DPARSF) (Yan & Zang, 2010). To ensure the reliabilty of functional data, we abandoned the first 10 volumes of images. Slice timing and head motion have been done in the remaining 230 volumes of images. No subject had a head movement bigger than 1.5 mm or rotation larger than 1.5°. The images were then normalized to the standard echo planar imaging (EPI) template (resampled voxel size: 3 mm × 3 mm × 3 mm). The following images were smoothed (full-width at half-maximum Gaussian kernel: 4 mm). After normalization, the time series was linearly detrended. Except global signal, 24 parameters of head motion (Friston et al., 1996), signals of white matter, and signals of cerebrospinal fluid were all removed. ALFF/d-ALFF was based on the frequency spectrum of rs-fMRI signals.

Total gray matter (GM) was obtained with the VBM8 toolbox as elaborated in the earlier work (Ma et al., 2016; Ma et al., 2015).

Static ALFF computation

ALFF was calculated using DPARSF toolkit as used in prior research (Cheng et al., 2019; Luo et al., 2012). With the aid of fast Fourier transform, the time series was transformed from the time domain to a frequency domain, from which the power spectrum was achieved. With the power spectrum of each voxel from all subjects, the square root was collected at each frequency and then averaged in the region of 0.01–0.08 Hz (Guo et al., 2013). The square root obtained was known as the ALFF at the given voxel. We divided the ALFF by the global mean ALFF for standardization.

d-ALFF computation

d-ALFF was processed with Temporal Dynamic Analysis (TDA) toolkit which was dependent on DPABI (Yan et al., 2016). According to a former report (Sakoglu et al., 2010), the window length was supposed to be sufficiently short to capture transient signals and long enough to detect slow changing signals. A sliding window with moderate size of 32 TR and a moving step length of 1 TR were selected in this study (Chen et al., 2019b). The 230 time points were divided into 199 windows. ALFF value was computed within each moving window for all participants. Then, the standard deviation (SD) of all ALFF maps from moving windows was computed to evaluate the variability of ALFF. Here, SD was used as d-ALFF.

Statistical analysis

Statistical analyses were processed with SPM12 toolkit. To compare differences in ALFF and d-ALFF of two groups, we employed two-sample t-tests method. The factors such as age, total GM volume, and gender were regressed. Gaussian Random Field (GRF) approach was adopted to perform multiple comparisons. The voxel level and the cluster level was set p < 0.01 and p < 0.05 respectively (the minimum cluster size in ALFF and d-ALFF analyses was 78 voxels) in the GRF correction.

Correlation analysis

Based on region of interest (ROI), Pearson’s correlation was analyzed to probe the relation of alterations in ALFF/d-ALFF to the clinical data of ALS. The mean ALFF/d-ALFF value of each significant clusters (ROIs) was used. A residual term was employed to correlate with clinical data. Meanwhile, the total GM volume, age, and gender were regressed. Bonferroni correction was introduced (significant level: p < 0.05/N) in the present study. Here, N = 15/12 represented the amount of comparisons using ALFF and d-ALFF.

Classification analysis

Support vector machine (SVM) method was utilized to compare classification ability among static ALFF, d-ALFF, and their combination for patients/HCs. The mean ALFF and d-ALFF of each static ALFF’s ROIs and d-ALFF’s ROIs were used as classification features. Liblinear toolbox with default parameter was utilized. Given that we aimed to compare the classification ability among ALFF, d-ALFF, and their combination, a leave-one-out cross validation (LOOCV) was accepted. LOOCV could obtain stable performance and prevent the possibility of overfitting (Chen et al., 2019a; Liu et al., 2015). There were m (m = 108) LOOCV loops. In each loop, we choose one participants’ information to test the categorization model and the m − 1 participants’ information was selected for model training. Finally, specificity, sensitivity and accuracy were collected to evaluate classifier performance.

Validation analysis

In order to confirm the main findings of d-ALFF, d-ALFF data with window lengths of 40 TRs and 50 TRs was recollected.

Results

Differences in static ALFF

The ALFF in the ALS group increased in the right inferior temporal gyrus, right medial superior frontal gyrus, and right medial superior frontal gyrus and reduced in the left middle occipital gyrus and left precentral gyrus. The details were available in Table 2 and Fig. 1A.

Table 2 Differences in ALFF between ALS and HC groups.

Clusters	Brain regions	Cluster size (voxels)	MNI (X, Y, Z)	T value	
ALS >HC	
Cluster 1	Right inferior temporal gyrus	128	45, −6, −45	4.14	
Cluster 2	Left superior orbitofrontal gyrus	442	−15, 18, −15	4.45	
Cluster 3	Right medial superior frontal gyrus	93	6, 54, 27	4.35	
ALS <HC	
Cluster 1	Left middle occipital gyrus	895	−27, 90, −3	−4.56	
Cluster 2	Left precentral gyrus	108	−27, −21, 72	−3.81	
Notes.

MNI, Montreal Neurological Institute. X, Y, Z, coordinates of primary peak locations in the MNI space.

ALS Amyotrophic Lateral Sclerosis

HC Healthy Control

T value denotes the statistic value of two-sample t-test by contrasting the ALS patients to the controls (p < 0.01, GRF-corrected at a cluster level of p < 0.05).

Figure 1 Results of ALFF and d-ALFF analyses by two-sample t-tests between ALS group and HC group.

(A) Brain regions with significant difference in static ALFF between the ALS group and HC group. (B) Brain regions with significant difference in d-ALFF between the ALS group and HC group. The voxel level was set at p < 0.01, and the cluster level was set at p < 0.05 with GRF corrected. The color bar represents the T value of the between-group analysis. Hot colors represent higher ALFF/d-ALFF in the ALS group than in the healthy control group, and cool colors represent the lower ALFF/d-ALFF in the ALS group than the healthy control group.

Differences in d-ALFF

As shown in Table 3 and Fig. 1B, d-ALFF did not increase in ALS group. Decreased d-ALFF was seen in the right lingual gyrus, left superior temporal gyrus, bilateral precentral gyrus, and left paracentral lobule.

Table 3 Differences in d-ALFF between ALS and HC groups.

Clusters	Brain regions	Cluster size (voxels)	MNI (X, Y, Z)	T value	
ALS >HC	
None					
ALS <HC	
Cluster 1	Right lingual gyrus	988	24,−45, −9	−4.35	
Cluster 2	Left superior temporal gyrus	176	−48,−27, 12	−3.82	
Cluster 3	Bilateral precentral gyrus	90	51,−9, 24	−3.45	
Cluster 4	Left paracentral lobule	125	−6, −15, 75	−3.70	
Notes.

MNI, Montreal Neurological Institute. X, Y, Z, coordinates of primary peak locations in the MNI space.

ALS Amyotrophic Lateral Sclerosis

HC Healthy Control

T value denotes the statistic value of two-sample t-test by contrasting the ALS patients to the controls (p < 0.01, GRF-corrected at a cluster level of p < 0.05).

Figure 2 Correlation between ALFF value in the left superior orbitofrontal gyrus and ALSFRS-R score in the ALS group.

Figure 3 Correlation between ALFF value in the left superior orbitofrontal gyrus and disease progression rate in the ALS group.

Correlation analysis

No correlation was detected between ALFF and clinical data in ALS. However, as shown in Fig. 2, the ALFF in the left superior orbitofrontal gyrus had a negative correlation with ALSFRS-R score at a trend level (p = 0.0096, r =  − 0.3495, uncorrected). The ALFF in the left superior orbitofrontal gyrus demonstrated a tendency of positive correlation with disease progression rate (Fig. 3, p = 0.0135, r = 0.3344, uncorrected). In addition, no significant difference between d-ALFF and clinical data was found.

Performance of classification

Figure 4 shows the receiver operating characteristic curve (ROC) with ALFF, d-ALFF, and their combination. The numerical data of the area under the curve (AUC) with ALFF, d-ALFF, and their combination were 0.82, 0.82 and 0.84 respectively.

Figure 4 Receiver operating characteristic curve of the classifier with ALFF, d-ALFF, and their combination.

As shown in Table 4, the ALFF method showed a classification accuracy of 76.85%, a specificity of 72.22%, and a sensitivity of 81.48%. The d-ALFF index exhibited a classification accuracy of 76.85%, a specificity of 62.96%, and a sensitivity of 90.74%. The accuracy, specificity, and sensitivity of the combined ALFF and d-ALFF were 79.63%, 72.22%, and 87.04%, respectively.

Table 4 Performance evaluation of classifier using ALFF, d-ALFF and combined ALFF and d-ALFF.

	Sensitivity (%)	Specificity (%)	Accuracy (%)	
ALFF	81.48	72.22	76.85	
d-ALFF	90.74	62.96	76.85	
Combined ALFF and d-ALFF	87.04	72.22	79.63	

Validation results

The results of d-ALFF using window sizes of 40 TRs and 50 TRs were very similar to the major results of 32 TRs. Validation results were available in Supplementary Materials (Figs. S1A and S1B).

Discussion

The present research studied the dynamic brain activity in ALS by using d-ALFF for the first time. We found that: (1) patients with ALS showed decreased d-ALFF in the right lingual gyrus, left superior temporal gyrus, bilateral precentral gyrus, and left paracentral lobule at resting state, and (2) the combined ALFF and d-ALFF distinguished ALS from HCs with higher accuracy than ALFF or d-ALFF alone.

Alterations in static ALFF

The brain areas with static ALFF differences in patients with ALS are consistent with previous reports (Luo et al., 2012; Ma et al., 2016), except that the right inferior temporal gyrus had increased ALFF. The temporal lobe with aberrant activation and connection in patients with ALS was discovered in preceding rs-fMRI articles (Li et al., 2018b; Loewe et al., 2017; Zhou et al., 2016). Besides, the thinning of cerebral cortex in the right inferior temporal gyrus is related to rapid clinical progression in ALS (Verstraete et al., 2012). The right inferior temporal gyrus is generally thought to be associated with social information processing for objects, places, and faces (Grill-Spector, 2003; Hall, Fussell & Summerfield, 2005). Of note, the cognitive impairment of ALS includes deficits in social cognition and executive functions (Beeldman et al., 2016). Moreover, deficits in recognition of facial expressions of emotion in ALS have been documented (Zimmerman et al., 2007). However, the change in the ALFF value was not found in previous research on ALS (Luo et al., 2012). The finding was probably caused by the situation that patients from the two studies were at different stages of the disease. The previous study recruited patients with an earlier stage compared with that in the present study, where the amplitude feature in this area may not altered. However, more longitudinal research is need to make the result clear. Therefore, this functional alteration may be the imaging evidence for understanding the impaired recognition of emotional stimuli in ALS at a certain stage. Elevated ALFF in the left superior orbitofrontal gyrus was relevant to rate of disease progression and ALSFRS-R score at a trend level. Hence, increased ALFF in this area might be useful to understand the progress of ALS.

Alterations in d-ALFF

Compared with HCs, the d-ALFF in the right lingual gyrus was lower in ALS group. Dysfunction of the right lingual gyrus in ALS was documented, including metabolic difference (Verma et al., 2013) and functional connectivity (Li et al., 2018b). The lingual gyrus is an important area in the visual system (Yang et al., 2015), and 24.13% of Chinese ALS population are considered with visuospatial disability (Wei et al., 2015). Thus, we concluded that impairment in the right lingual gyrus over time might underlie the phenomenon of visual dysfunction in ALS.

This study also found reduced d-ALFF in the left superior temporal gyrus in ALS group. The abnormality in left superior temporal gyrus was same with previous fMRI studies in regional functional connectivity density (Li et al., 2018b) and with anatomical MRI studies in gray matter volume (Buhour et al., 2017; Kim et al., 2017; Sheng et al., 2015). Electroencephalography study (McMackin et al., 2019) shows decreased power in the left superior temporal gyrus when patients with ALS underwent auditory frequency-mismatch oddball paradigm. The left superior temporal gyrus was considered to be related to the function of auditory working memory (Leff et al., 2009). The quieter activity in the left superior temporal gyrus over time at resting state in ALS can be explained as the reason of memory decline in ALS.

We also observed decreased d-ALFF in the bilateral precentral gyrus and left paracentral lobule in ALS. These motor regions are hallmark areas for patients with ALS who had structural (Cosottini et al., 2012; Schmidt et al., 2014; Thorns et al., 2013) and functional (Ma et al., 2016; Zhang et al., 2017; Zhou et al., 2014) abnormalities. These motor regions were detected with static ALFF and d-ALFF indices in the current research, indicating the vital role of these regions in studying ALS.

Relationship between static ALFF and d-ALFF changes

d-ALFF and ALFF detected decreased activity in the precentral gyrus in patients suffering from ALS. These findings provide a helpful perspective for our understanding the motor neuron dysfunction of this disease. In addition, d-ALFF could provide other different changes compared with traditional ALFF method, showing that dynamic brain activity may be an important neuroimaging feature to track pathological changes in ALS.

Altered d-ALFF could identify patients with ALS from HCs, and the classification performance is similar to that of ALFF. However, when both static and dynamic ALFF features were combined, the classification performance achieved the highest overall accuracy rate. These results consolidated that ALFF and d-ALFF were different approaches used to characterize brain activity from different perspectives. In contrast to ALFF, d-ALFF could provide complementary information to understand ALS better. The findings also provided a novel way to help distinguish patients with ALS from the healthy population.

Limitations and further considerations

Several limitations should be noted in this work. First, the features in classification were based on prior knowledge, which may increase the overall accuracy rate. The combined ALFF and d-ALFF approach would enhance accuracy with a single feature. More subjects and further sub-group analysis should be considered to obtain stable and more precise results.

Conclusions

ALFF and d-ALFF patterns were altered in patients with ALS. The alterations in the two features could identify ALS at the individual level with nearly the same performance. However, when the two features were combined, the classification performance achieved the highest overall accuracy rate. These results provide evidence for applying dynamic spontaneous neural activity (d-ALFF) to uncover the neuropathology of ALS.

Supplemental Information

Data S1 Raw data and software for img format

Data imagings (T map), subject information and software for img format (xjview).

Click here for additional data file.

Figure S1 Results of d-ALFF analysis by two-sample t-test between ALS group and HC group

a. Results of the sliding-window length of 40 TR. b. Results of the sliding-window length of 50 TR.

Click here for additional data file.

Additional Information and Declarations

Competing Interests

Author Contributions

Human Ethics

Data Availability

The authors declare there are no competing interests.

Xujing Ma and Fengmei Lu analyzed the data, prepared figures and/or tables, authored or reviewed drafts of the paper, and approved the final draft.

Heng Chen, Caihong Hu, Jiao Wang, Sheng Zhang, Shuqin Zhang and Guiran Yang analyzed the data, authored or reviewed drafts of the paper, and approved the final draft.

Jiuquan Zhang conceived and designed the experiments, performed the experiments, authored or reviewed drafts of the paper, and approved the final draft.

The following information was supplied relating to ethical approvals (i.e., approving body and any reference numbers):

This study was approved by the medical research ethics committee of Southwest Hospital (The First Affiliated Hospital of the Third Military Medical University of the Chinese People’s Liberation Army).

The following information was supplied regarding data availability:

The raw data are available in the Supplemental Files.

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
