# Peer review of "Static and dynamic alterations in the amplitude of low-frequency fluctuation in patients with amyotrophic lateral sclerosis"

_PeerJ, doi:10.7717/peerj.10052_

## Round 0.1 · original submission · Major Revisions

All the reviewers raised major concerns, and the authors should carefully consider and address these comments.

Reviewer 1 ·

Basic reporting

no comment

Experimental design

no comment

Validity of the findings

no comment

Additional comments

The authors investigated ALFF and d-ALFF alterations in ALS patients, and found some interesting results. The ALS patients can be more accurately classified from HC by combining ALFF and d-ALFF features compared with any single feature.
1. There are many grammar errors and confusing sentences in the whole manuscript. E.g. what does the subtitles (comparison of ALFF, comparison of d-ALFF, comparison of ALFF and d-ALFF, comparison of brain regions) mean? What does the “unusual areas” mean? They should check the whole manuscript, revise those sentences carefully and make it be understood more easily. I suggest the authors ask a native English speaker or company with English editing for help.
2. According to the title, it seems that the d-ALFF is more appropriate than ALFF in understanding mechanism of ALS. But the article told us that combination of ALFF and d-ALFF is better. So the authors should discuss what the advantage of d-ALFF is for understanding mechanism of ALS, or maybe the title “Characterizing stationary and dynamic amplitude of low-frequency fluctuation in ALS” is more appropriate.
3. In section of Data acquisition, the voxel size is 3x3x4, matrix size is 64x64, and the data was scanned axially. So the thickness would be 4 mm instead of 3 mm. The authors should check the information more carefully.
4. What is the resampled voxel size in the normalized images?
5. In Table 1. There should be p-value for the comparison of gender. Also it’s better to include GM information in Table 1.
6. Since there was no difference in gender and age, why the authors regressed out them in statistical analysis
7. There are some errors in the sentence “As shown in Table 2 and Figure 1, the ALFF in ALS patients increased in right inferior temporal gyrus, right medial superior frontal gyrus and right medial superior frontal gyrus”
8. In table 4, the authors included brain areas to movement, visual, memory, emotion or cognition network. How did they do the work? Is there any atlas used?
9. “We have investigated whether d-ALFF is able to discriminate ALS patients from healthy control” and “In addition, the classification performance with ALFF, d-ALFF and combined ALFF and d ALFF were compared”. The two sentences were overlapped and should be revised.
10. In discussion, the word “hyperactivity” is not correct because this is a resting-state study instead of task study.
11. Discussing the result of significant correlation between ALFF and ALSFRS-R score, disease progression rate

Reviewer 2 ·

Basic reporting

No comment.

Experimental design

No comment.

Validity of the findings

No comment.

Annotated reviews are not available for download in order to protect the identity of reviewers who chose to remain anonymous.

·

Basic reporting

1. The English language should be improved to ensure that an international audience can clearly understand your text. Some examples where the language could be improved include lines 44-46 – ALS is a neurodegenerative disorder manifested as cognitive and motor impairments, typically resulting in death from…...
2. The introduction is weak. A lot of details have been given as for the application of d-ALFF while no efforts have been spent to provide a clear hypothesis and rationale of the study. A paragraph reporting study hypothesis should be included.

Experimental design

3. Both DPARSF and DPABI appeared in the Methods, have you used both of them or only DPABI?
4. AlphaSim correction is not suggested. After AlphaSim was questioned by a previous study, it’s correction of multiple comparisons updated with a statistical threshold of P < .050 (an uncorrected height threshold of P < .001, in which the z statistical image was used to estimate smoothness) based on the modified version of DPABI.
5. Why EPI template and not DARTEL?
6. In the correlation analysis, the correction of multiple comparisons is needed.
7. In Classification analysis, how many features were used and what are they? Please clarify. No rigorous validation (e.g., cross validation of K-fold cross validation) is employed. And strictly, there must be one-fold data that is never used in any training process. Is it true in the present work? More details are deeded to address those questions.

Validity of the findings

no comment

Additional comments

9. The discussion section is very weak and lacks the explanation/justification and or discussion of the obtained results.

---

## Round 0.2 · accepted · Accept

All the reviewers have accepted the manuscript.

Reviewer 1 ·

Basic reporting

no comment

Experimental design

no comment

Validity of the findings

no comment

Additional comments

I am satisfied with the revision , and suggest acceptance of the paper

Reviewer 2 ·

Basic reporting

No comment.

Experimental design

No comment.

Validity of the findings

No comment.

Additional comments

All my concerns have been addressed. I am satisfied with the revision and this manuscript can be accepted.

·

Basic reporting

no comment

Experimental design

no comment

Validity of the findings

no comment

Additional comments

This article is innovative and I hope the author will have better papers to publish in the future.